# Post-Vaccination Anti-SARS-CoV-2-Antibody Response in Patients with Multiple Myeloma Correlates with Low CD19+ B-Lymphocyte Count and Anti-CD38 Treatment

**DOI:** 10.3390/cancers13153800

**Published:** 2021-07-28

**Authors:** Susanne Ghandili, Martin Schönlein, Marc Lütgehetmann, Julian Schulze zur Wiesch, Heiko Becher, Carsten Bokemeyer, Marianne Sinn, Katja C. Weisel, Lisa B. Leypoldt

**Affiliations:** 1University Cancer Center Hamburg, Department of Oncology, Hematology and Bone Marrow Transplantation, Section of Pneumology, University Medical Center Hamburg-Eppendorf, Martinistraße 52, 20246 Hamburg, Germany; m.schoenlein@uke.de (M.S.); c.bokemeyer@uke.de (C.B.); ma.sinn@uke.de (M.S.); k.weisel@uke.de (K.C.W.); l.leypoldt@uke.de (L.B.L.); 2The Institute of Medical Microbiology, Virology and Hygiene, University Medical Center Hamburg-Eppendorf, Martinistraße 52, 20246 Hamburg, Germany; mluetgehetmann@uke.de; 3Department of Internal Medicine, Division of Infectious Diseases, University Medical Center Hamburg-Eppendorf, Martinistraße 52, 20246 Hamburg, Germany; j.schulze-zur-wiesch@uke.de; 4Institute for Medical Biometry and Epidemiology, University Medical Center Hamburg-Eppendorf, Martinistraße 52, 20246 Hamburg, Germany; h.becher@uke.de

**Keywords:** multiple myeloma, anti-SARS-CoV-2 antibody response, SARS-CoV-2 vaccination

## Abstract

**Simple Summary:**

The global impact of the current COVID-19 pandemic has led to the impressively rapid development of multiple anti-SARS-CoV-2 vaccines. However, only few data are available regarding the efficacy of anti-SARS-CoV-2 vaccines in patients with hematological malignancies, and, in particular, plasma cell neoplasia. This ongoing observational study aimed to describe the level of post-vaccination anti-SARS-CoV-2-antibodies depending on multiple clinical factors including B lymphocyte count and current therapy of 82 patients with multiple myeloma and related plasma cell neoplasia, after the first dose of anti-SARS-CoV-2 vaccination. A positive SARS-CoV-2 spike protein antibody titer (SP-AbT) was detected in 23% of assessable patients. SARS-CoV-2 SP-AbT was significantly higher in patients with higher CD19+ B lymphocyte counts and current treatment with anti-CD38-antibodies has led to significantly reduced SP-AbT titers. Based on our results, the majority of myeloma patients respond poorly after receiving the first dose of any anti-SARS-CoV-2 vaccination and need booster vaccination.

**Abstract:**

Few data are available regarding the efficacy of anti-SARS-CoV-2 vaccines in patients with hematological malignancies, and particular, plasma cell neoplasia. This ongoing single-center study aimed to describe the level of post-vaccination anti-SARS-CoV-2-antibodies depending on B lymphocyte count, current therapy, and remission status of patients with multiple myeloma and related plasma cell dyscrasia, after the first dose of anti-SARS-CoV-2 vaccination. The 82 patients included in this study received SARS-CoV-2 vaccines (including mRNA- and vector-based vaccines) as a routine measure. After the first vaccination, a positive SARS-CoV-2 spike protein antibody titer (SP-AbT) was detected in 23% of assessable patients. SARS-CoV-2 SP-AbT was significantly higher in patients with higher CD19+ B lymphocyte counts. A cut-off value of ≥30 CD19+ B cells/µL was significantly positive correlating with higher SARS-CoV-2 SP-AbT. In contrast, current treatment with anti-CD38-antibodies has led to significantly reduced SP-AbT titers. Furthermore, in multivariable linear regression, higher age and insufficiently controlled disease significantly correlated negatively with SARS-CoV-2 SP-AbT. Conversely, treatment with immunomodulatory drugs did not harm the development of antibody titers. Based on our results, the majority of myeloma patients respond poorly after receiving the first dose of any anti-SARS-CoV-2 vaccination and need booster vaccination.

## 1. Introduction

Infections, regardless of whether they are bacterial or viral in origin, pose a major threat to patients with multiple myeloma (MM), due to plasma cell dysfunction-related immunodeficiency [1]. Patients with newly diagnosed, relapsed, or refractory (RR) MM regularly show decreased numbers of CD19+ B-lymphocytes and lower levels of polyclonal immunoglobulins, the latter being related to a suppression of CD19+ B-lymphocytes [2]. Even though MM is not typically associated with a clinical T-cell immunodeficiency state, both quantitative and functional T-cell abnormalities have previously been described [3,4,5]. In addition, anti-myeloma treatments, and in particular the use of anti-CD38-monoclonal antibodies like daratumumab and isatuximab, are associated with impaired immune response. Due to specific targeting, treatment with anti-CD38-monoclonal antibodies leads to long and sustained suppression of CD38+ plasma cells [6]. However, the expression of CD38+ is not limited to plasma cells but also found on various immune cells including B lymphocytes [7,8]. More specifically, B lymphocyte precursor cells show high expression of CD38 and are thus also being targeted by anti-CD38-directed treatment [8,9].

During the current COVID-19 pandemic, an increased COVID-19-related overall mortality for MM patients has been reported in general, with a rate of 33% in a large international registry [10]. However, the global impact of the current COVID-19 pandemic has led to the impressively rapid development of multiple effective anti-SARS-CoV-2 vaccines. In December 2020, the first- and second-in-class COVID-19 messenger ribonucleic acid (mRNA) vaccines BNT162b2 and mRNA1273 were authorized by the US Food and Drug Administration for Emergency Use Authorization for the prevention of COVID-19 and shortly after also by the European Medicines Agency (EMA) [11,12,13,14]. Additionally, in January 2021, the EMA recommended the authorization of the first-in-class anti-SARS-CoV-2 vector vaccine AZD1222 in the European Union [15].

Based on high COVID-19-associated mortality in cancer patients in general and for patients with hematological malignancies in particular, both the American Society of Hematology and the European Hematology Association recommended the prioritization of patients with hematological malignancies including patients with MM for anti-SARS-CoV-2 vaccination [16,17]. The International Myeloma Society (IMS) recommends the immunization of all patients with MM or precursor diseases; in short, treatment should not be interrupted due to vaccination in situations of active progressive myeloma [18]. However, reliable results regarding the efficacy of anti-SARS-CoV-2 vaccines in patients with untreated or treated MM and especially under the influence of proteasome inhibitors (PI), immunomodulatory drugs (IMiD), and anti-CD38 or anti-SLAMF7 monoclonal antibodies are scarcely available. Therefore, this single-center study aimed to describe the level of post-vaccination anti-SARS-CoV-2-antibodies depending on therapy, remission status, and B- and T-cell numbers in patients with MM and related plasma cell neoplasia including monoclonal gammopathies of clinical significance (MGCS) and systemic light-chain amyloidosis (AL). Here, we report an analysis regarding post-vaccination antibody titers after the first vaccination.

## 2. Materials and Methods

### 2.1. Study Design and Patients

In this observational single-center study patients aged 18 years and older with a confirmed diagnosis of MM, MGCS, or AL according to the 2014 updated diagnostic criteria of the international myeloma working group (IMWG) and who were vaccinated according to national prioritization strategy and eligible for Anti-SARS-CoV-2 vaccination according to IMS recommendations were included [18,19]. This single-center analysis was performed between 1 January and 21 May 2021, at the Department of Oncology and Hematology at the University Medical Center Hamburg-Eppendorf, Germany. Exclusion criteria were prior confirmed infection with SARS-CoV-2 defined as either (a) polymerase-chain-test-validated infection or (b) evidence of anti-SARS-CoV-2 nucleocapsid antibodies before immunizations.

The primary aim of this study was to evaluate a possible correlation between antibody titers and CD19+ B lymphocyte count. Secondly, we investigated the number of patients with plasma cell neoplasia showing anti-SARS-CoV-2-antibodies at least seven days after their first immunization and the development of anti-SARS-CoV-2-antibodies under CD38-directed treatment and their relationship to remission status, disease, and patient characteristics, as well as the relationship between antibody titer and B and T cell status.

Immunizations themselves were no part of this purely observational study. They were organized by patients and done according to recommendations and prioritization of the Standing Committee on Vaccination at the Robert Koch Institute by their primary care physicians or at vaccination centers.

All vaccines currently available in Germany could be used. These were Comirnaty^®^ (previously: BNT162b2 by BioNTech, Mainz, Germany; Pfizer, New York, NY, USA), and Moderna vaccine (previously: mRNA-1273 by Moderna, Cambridge, MA, USA), summarized as mRNA-vaccines, and Vaxzevria^®^ (previously: AZD1222 by AstraZeneca, Oxford, UK), a vector-based vaccine. No data were collected regarding vaccination side effects or toxicities.

Myeloma-directed treatment of patients under active therapy was adapted according to the recommendations of the IMS in routine clinical care [18].

Clinical data regarding treatment and disease characterization were collected from the patients’ electronic medical records. For assessment of patient comorbidities, the Charlson Comorbidity Score was used [20]. Since multiple myeloma itself is defined as a distinct group of non-Hodgkin B-Cell lymphoma, all patients included in this study achieved at least two points in this score.

Remission status for multiple myeloma was performed according to the 2016 updated IMWG consensus criteria for response and minimal residual disease assessment in MM [21]. For statistical analysis, groups of well-controlled disease status (≥very good partial remission (VGPR)) and insufficiently controlled disease (≤partial remission) at the time of vaccination were defined. AL remission status was assessed by corresponding consensus criteria [22,23]. Treatment lines were counted as recommended by Rajkumar et al. [24]; patients in first-line treatment including maintenance therapy were counted as a newly diagnosed disease. Triple- and penta-refractory disease were defined similarly to Gandhi et al. [25].

This study was part of the COVIDOUT trial registered at ClinicalTrials.gov (ClinicalTrials.gov Identifier: NCT04779346) and was approved by the Ethics Committee of the Medical Council of Hamburg (reference 2020-10275-BO-ff). Written informed consent was provided by each patient according to local requirements.

### 2.2. Detection of Anti-SARS CoV-2 Antibodies

Samples were analyzed for the presence of SARS-CoV-2-specific antibodies using the quantitative anti-spike IgG (LIAISON SARS-CoV-2 spike trimeric IgG; DiaSorin, Saluggia, Italy, cut off ≥ 33.8 BAU/mL) and qualitative anti-NC Ig assay (Elecsys Anti-SARS-CoV-2, Roche; cut off ≥ 1 COI). CLIAs were performed using the immuno-analyzer (Cobas e411, Roche, Mannheim, Germany; and Liaison XL, DiaSorin, Saluggia, Italy) according to the manufacturer’s recommendations.

### 2.3. Flow Cytometry Procedure

Flow cytometric analyses were performed for assessment of the patients’ lymphocyte status on a Navios flow cytometer with CXP Software (Beckmann Coulter Krefeld, Germany). Patients’ blood samples were collected in EDTA tubes and stained according to local standard protocol routine using antibodies for CD45, CD4, CD8, and CD3 (article no. 6607013) to identify T cell subsets and for CD45, CD56, CD19, CD16, and CD3 (article no. 6607073) to identify B cells and NK cells (all antibodies by Beckmann Coulter Krefeld, Germany).

### 2.4. Statistical Analysis

The relationships between antibody titers and variables of interest (B lymphocyte count, duration of CD38-directed treatment and demographic and clinical variables) were evaluated by correlation analysis and multivariable linear regression. In the linear regression the joint effect of CD19 count, age, sex, time since vaccination, controlled disease (no remission), and type of vaccine on the value of the antibody titer was investigated. Backward elimination was used for model selection. The selected model was repeated including individuals with positive antibody titer only. Due to the highly skewed distribution of CD19 count and antibody titer, both variables were log-transformed for linear regression analysis. In addition, a logistic regression model was fitted with positive antibody titer (cut off ≥ 33.8 BAU/mL) as binary outcome. Comparisons between groups were performed by Mann–Whitney U test for continuous and by Fisher’s exact test for categorical characteristics. *p*-value < 0.05 was considered as statistically significant. The reported *p*-values are two-tailed. All statistical analyses were performed, and figures were designed, by using the Statistical Package for Social Sciences statistical software, version 26.0 (IBM Corp., Armonk, New York, NY, USA) and by SAS software (version 9.4 of the SAS System for Windows; SAS Institute, Inc, Cary, NC, USA).

## 3. Results

A total of 82 patients were included in this observational trial (Figure 1).

Patient demographics and characteristics are presented in Table 1.

The median age was 67.5 years (range 40–85) years. A total of 78 patients had MM, 2 MGCS, and 2 AL. Of these patients, 40 had newly diagnosed disease (48.8%) with 23 of those receiving maintenance treatment. Of the remaining 42 patients with RR disease, 10 were triple- and 3 penta-refractory and 11 patients (13.4%) had had more than three prior treatments. At the time of vaccination, 69 patients (84.1%) were receiving anti-myeloma treatment. In total, 34 patients received anti-CD38- and 3 anti-SLAMF7-targeting therapies and 52 patients received IMiD-based treatments (thalidomide: 1, lenalidomide: 45, pomalidomide: 6); 69.5% of all patients achieved well-controlled disease status (≥VGPR).

A total of 63 patients was vaccinated with mRNA-based and 19 with vector-based vaccines, respectively (Table 2). Assessment of anti-SARS-CoV-2 antibody titers took place on a median of 25 days after the first vaccination (25.2 (±11.8)). For further evaluation of the immune status, flowcytometric analysis was performed as part of the clinical routine and revealed a mean CD19+ B lymphocyte count of 88.5 cells/µL (range 1–642/µL), a CD4+ T cell count of 442.7/µL (range 39–1730/µL), and 553.4/µL CD8+ T cells (range 27–2261/µL).

Across the complete cohort, the antibody titer (spike trimer) after the first vaccination was at a median of 6.9 BAU/mL (mean: 138.1; range: 0–3640). In this cohort 57 of 74 assessable patients (77.0%) had negative and 17 (23%) positive test results (8 patients’ results were not assessable).

To further evaluate underlying factors impacting vaccination response, we first assessed a possible relationship between antibody titers and CD19+ B-lymphocyte counts. We found a significant positive correlation between higher CD19+ B lymphocyte counts and the SARS-CoV-2 spike protein antibody titer (SP-AbT) result (Spearman correlation coefficient: *r* = 0.45; *p* < 0.0001). Individuals with current anti-CD38-antibody treatment showed lower antibody titers (*p* = 0.009, Mann–Whitney U-Test) and the likelihood for positive titer results was significantly lower in patients with current anti-CD38 treatment compared to those without (ongoing anti-CD38 treatment: negative titer 29 (93.5%), positive titer 2 (6.5%); without anti-CD38 treatment: negative titer 28 (65.1%), positive 15 (34.9%): *p* = 0.005 Fisher’s exact test) (Figure 2A). Under ongoing anti-CD38 treatment, there were no differences in antibody titer results between daratumumab- or isatuximab- treated patients. Also, there was no correlation of current treatment with IMiDs.

As higher age is known to impact immune responses, we also assessed a possible relationship between higher age and antibody titers. A clear trend towards impacted antibody titers was seen with rising age (*r* = −0.30; *p* = 0.0093). In linear regression (Table 3), we found CD19+ B lymphocyte counts (logarithmic), age, time since vaccination, and insufficiently controlled disease significantly related to the height of the antibody titer level (R2 = 0.33) (Figure 2A–E). The result was similar when including observations with positive antibody titer only, except for a reduced nonsignificant effect for insufficiently controlled disease.

No significant effects in the regression were observed for the type of vaccine (*p* = 0.60) (Figure 2D). The result was similar when including observations with positive antibody titer only, except for a reduced nonsignificant effect for insufficiently controlled disease. In the multivariable logistic regression model we observe odds ratios (OR) with 95% confidence intervals (CI) as follows: CD19+ B lymphocyte counts (logarithmic) OR = 1.76, CI 1.11–2.81, age: OR = 0.91, CI:0.84, 0.97; time since vaccination: 1.04, CI:0.99–1.10; and insufficiently controlled disease OR = 0.36, CI:0.06–2.24, thus paralleling the results of the linear regression analysis.

Next, we assessed possible reasons and impacting factors on CD19+ B lymphocyte counts. Median numbers of CD19+ B lymphocytes were lower in patients under current anti-CD38-directed treatment compared to patients without treatment (median 20 vs. 38 CD19+ B lymphocytes, *p* = 0.054) although the difference was not statistically significant.

A similar statistical trend was observed for a correlation of CD19+ B lymphocyte counts and the duration of anti-CD38-directed treatment with longer durations of CD38 treatment leading to lower B cell counts (correlation coefficient *r* = −0.173; *p* = 0.06).

Next, we defined two groups of patients according to their CD19+ B lymphocyte counts (high vs. low); since the median number of CD19+ B lymphocytes was 28.5/µL, a cut-off value of ≥ 30/µL was chosen (high). For the high CD19+ B lymphocyte group, a significant positive correlation with higher anti-SARS-CoV-2 SP-AbT could be seen (*p* = 0.015), and median numbers of antibody titers were significantly higher in the CD19+ B lymphocyte high group (median: 16.16 BAU/mL vs. 0 BAU/mL, *p* < 0.0005; mean: 277.8 BAU/mL vs. 19.4 BAU/mL, Figure 3).

Since the process of antibody formation depends on the interaction of B and T lymphocytes, we also assessed the impact of CD4+ and CD8+ lymphocyte counts on the development of antibody titers. Indeed, we found a positive correlation between CD4+ lymphocyte counts (Spearman 0.21; *p* = 0.069), and a negative correlation with CD8+ T lymphocyte counts (−0.20; *p* = 0.087).

Of note, one patient of the overall cohort developed COVID-19 confirmed by highly positive PCR (95 × 106/copies per mL) on day 25 after her first vaccination. At this time point, the patient had no antibody titer while subsequently after infection, she eventually became positive for both nucleocapsid and spike protein antibody.

## 4. Discussion

This observational study aimed to describe the level of post-vaccination anti-SARS-CoV-2-antibodies depending on B lymphocyte count, current therapy, remission status, and host factors in patients with MM or related plasma cell neoplasia. As part of our analysis after the first vaccination, a positive anti-SARS-CoV-2 SP-AbT was seen in 23% of patients while the majority (77%) did not develop detectable antibody titers. In a comparison of cancer patients to healthy controls after the first vaccination with BNT162b2, Monin-Aldama et al. described rates of seroconversion of 97, 39, and 13% in healthy controls, solid cancer patients, and patients with hematological malignancies, respectively [26]. Indeed, high seroconversion rates of about 95% for healthy individuals after the first vaccination have been confirmed by others across different vaccine types, but with lower antibody titers in older individuals [27,28,29,30].

In further analyses, we found a significantly higher anti-SARS-CoV-2 SP-AbT in patients with higher CD19+ B cell counts. In contrast, current treatment with anti-CD38-antibodies led to significantly reduced antibody titers. In addition, there was a significant correlation of lower antibody titers with higher age and insufficiently controlled disease. Current treatment with IMiDs did not have any negative impact on the development of anti-SARS-CoV-2 SP-AbT. A positive correlation between CD4+ T lymphocyte counts and the antibody titer was observed while CD8+ T lymphocyte counts were negatively correlated.

Our results are in line with previously reported studies showing an impaired development of anti-SARS-CoV-2 antibodies in patients with MM and other plasma-cell-related neoplasia [26,31,32]. Terpos et al. investigated the level of anti-SARS-CoV-2 neutralizing antibody titers 22 days after the first vaccination with BNT162b2 in 48 elderly patients with MM or other plasma-cell-related neoplasia (however with >25 % patients without current anti-myeloma treatment) compared to a healthy control group of same median age. Here, significantly lower nAb titers in patients with plasma-cell-related neoplasia compared to the control group (25.0% vs. 54.8%) were observed [33]. Similarly, Bird et al. investigated the level of SARS-CoV-2 SP-AbT in 93 patients with MM after one dose of either BNT162b2 or ChAdOx1. At least 21 days after the first vaccination (median of 33 days), anti-SARS-CoV-2 SP-AbT were detectable in 59.1% of patients, the higher proportion observed here may in part be due to a later timepoint of response assessment after the first vaccination and to the lower percentage of patients under current anti-CD38-directed therapy. No differences between vaccine types were documented. Besides younger age as a factor associated with a positive immune response after the first vaccination, Bird et al. observed three other correlating factors: well-controlled disease status, absence of immune paresis, and the number of previous lines of therapy. Furthermore, being on any treatment was associated with a lower titer [32]. Nonetheless, younger age as a negatively impacting factor is not restricted to hematological patients but is also observed in healthy individuals [29,30]. Our results are paralleled by those currently reported by Pimpinelli et al. who observed lower levels of anti-SARS-CoV-2 SP-AbT in 42 patients with MM (median age 73 years) who were vaccinated twice with BNT162b2. After the first vaccination, 21.4% achieved a positive titer, compared to 52.8% in the control group. However, the proportion of MM patients with a positive anti-SARS-CoV-2 SP-AbT (defined as a cutoff >15 AU/mL) increased up to 78.6 % two weeks after the second vaccination. Univariate analysis regarding the likelihood of response after completed vaccination revealed a significant association with the type of current treatment. MM patients with PI- or IMiD-based therapies (alone or in combination) without daratumumab had a significantly higher likelihood of response compared to those who received an active daratumumab-based therapy (92.9% vs. 50%, *p* = 0.003) [31].

In contrast to the above-mentioned data, in our analysis, no significant differences in antibody titers of patients were found regarding the number of prior therapies but trends could be seen. The main factors associated with poor immune response were age and remission status which is in line with reported data and current treatment with CD38-antibody. In addition, we could show for the first time that lower CD19+ B lymphocyte counts significantly correlate with poorer antibody response to vaccination which may become a predictor of vaccination response.

Albeit this was a prospectively aimed observational trial, there are some limitations. Based on the comparatively small cohort size and the use of different anti-SARS-CoV-2 vaccines, there is a risk of potential bias and residual confoundings. In addition, since assessments were performed as part of the clinical routine in a real-world setting, the timepoint of anti-SARS-CoV-2 SP-AbT measurement after the first vaccination was not standardized and took place within a range of 7–63 days. Thus, in 11 of all assessable patients, SP-AbT were tested before day 14. This may have partially led to lower antibody titers and will be important to follow up in further analyses. Furthermore, observations regarding T-cell status only included numbers but no assessment of the induction and magnitude of a SARS-CoV-2 spike-specific T-cell response.

## 5. Conclusions

To the best of our knowledge, this is the first analysis of SARS-CoV-2 SP-AbT in patients with plasma-cell-related neoplasia with a systematic assessment of immune status including CD19+ B lymphocyte counts. As we have been able to demonstrate for the first time that lower CD19+ B lymphocyte counts significantly correlate with poorer SARS-CoV-2 SP-AbT we conclude that CD19+ B lymphocytes are essential for the development of a vaccination response with a suggested cut-off value of 30 CD19+ cells/µL to predict a sufficient immune response. In line with prior results, CD38-depleting therapies significantly impair antibody titer development. However, due to the long half-life of CD38-antibodies, intermittent pausing of these therapeutics for several months as would be needed to fully clear the antibody seems impractical and would likely endanger successful myeloma treatment. In addition, in line with recommendations of the IMS, concurrent treatment with IMiDs does not impair vaccination response and may thus be continued throughout the vaccination process.

Based on the results of our current analysis, the follow-up data are eagerly awaited to investigate if a boost reaction by a second vaccination will lead to a reliable improvement of anti-SARS-CoV-2 antibody response as previously described by Pimpinelli et al. and to evaluate the predictive value of CD19+ B lymphocyte count for antibody titers. In patients under current CD38-directed treatment, response assessment after vaccination should be mandatory. Additionally, patients need to be informed and should be advised to continue protective measures (e.g., masks and social distancing) due to the high risk of insufficient vaccination response. Since a relevant proportion of patients could remain insufficiently unprotected, the vaccination of family members and close contacts is essential for the development of herd immunity and thereby protection of patients at risk.

## Figures and Tables

**Figure 1 cancers-13-03800-f001:**
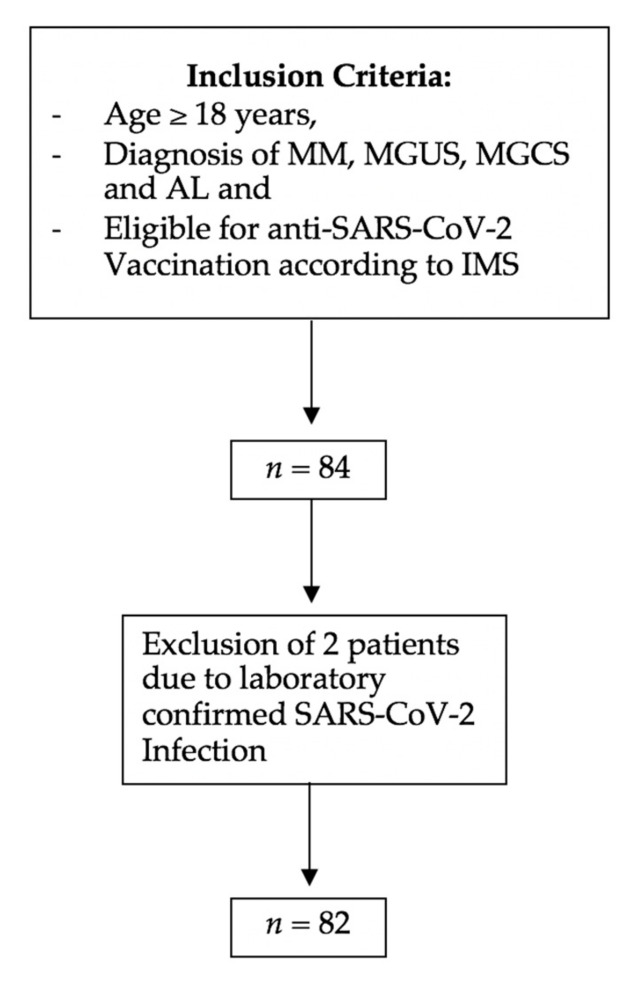
Flow chart of inclusion and exclusion criteria. MM, multiple myeloma; MGCS, monoclonal gammopathy of clinical significance; MGUS, monoclonal gammopathy of unknown significance; AL, systemic light-chain amyloidosis.

**Figure 2 cancers-13-03800-f002:**
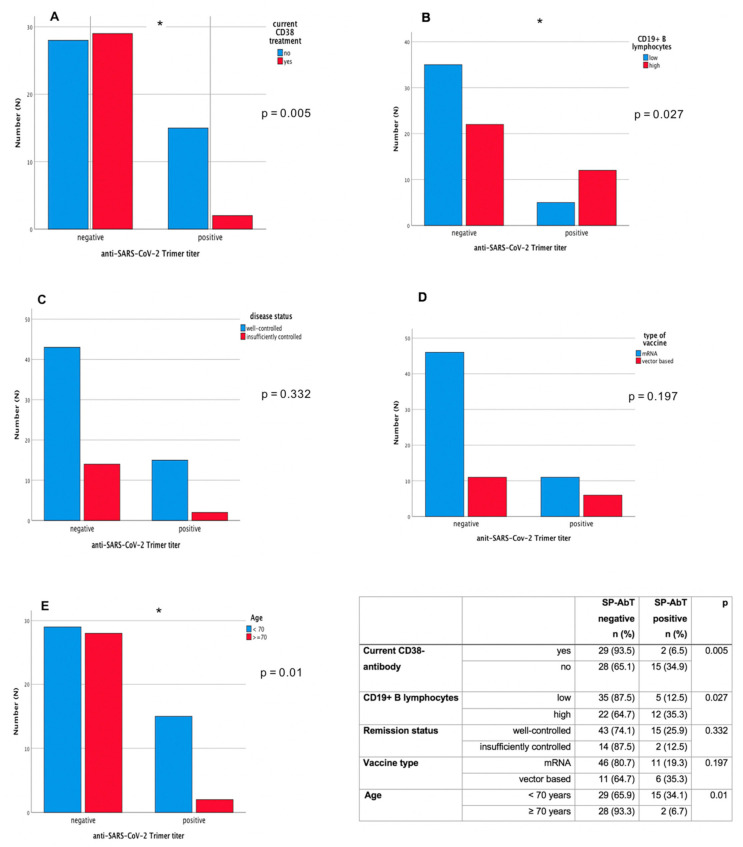
Anti-SARS-CoV-2-antibody response (anti spike-protein-trimer antibody titer) in patients with different characteristics. (**A**) Anti CD38-treatment, (**B**) CD19+ B lymphocyte count, (**C**) disease remission status, (**D**) vaccine type, and (**E**) age. * indicates *p* < 0.05.

**Figure 3 cancers-13-03800-f003:**
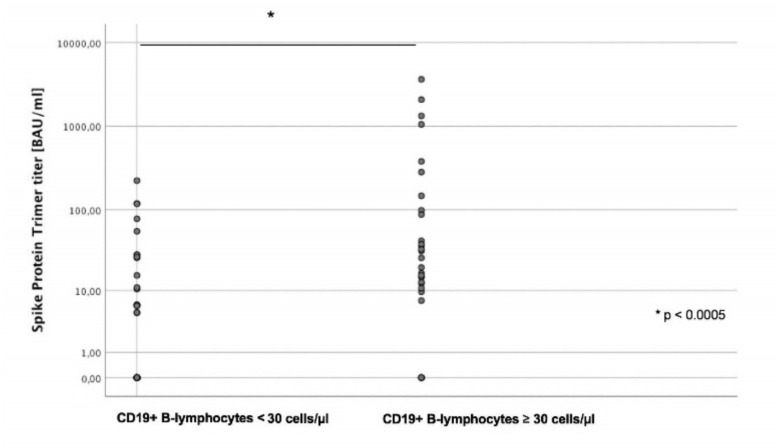
Median numbers of anti-SARS-CoV-2 antibody titers (spike-protein trimer) (Y axis, logarithmic) depending on CD19+ B lymphocyte count (X axis). Antibody titers were significantly lower in patients with low compared to high CD19+ B lymphocyte count (<30 vs. ≥30 cells/µL).

**Table 1 cancers-13-03800-t001:** Patient demographics and characteristics regarding plasma-cell-related disease and anti-myeloma treatment.

Variable	SP-AbT Negative	SP-AbT Positive	SP-AbT Missing	*p*-Value	Total
Age, median age in years (range)	68.54	58.65	68.5	<0.01	67.5 (40–85)
Sex, *n* (%)					
Male	33	12	4	0.41	49 (59.8)
Female	24	5	4		33 (40.2)
Charlson Comorbidity Index (mean; range)	2.94	2.56	2.75	0.01	2.72 (2–6)
Plasma-cell-related neoplasia, *n* (%)					
MM					78 (95.1)
Smoldering MM					0
MGCS					2 (2.4)
MGUS					0
AL					2 (2.4)
Newly diagnosed	25	11		0.17	40 (48.8)
R/R Disease	32	6			42 (51.2)
Triple-refractory	10	0			10 (12.2)
Penta-refractory	3	0			3 (3.7)
Anti-myeloma therapy, *n* (%)					69 (84.1)
Anti-CD38 monoclonal antibody in total	29	2	3	0.005	34 (41.5)
Daratumumab-based					25 (30.5)
Isatuximab-based					9 (11.0)
Mean duration of CD38-based therapy in months (range)					9.5 (0–50)
Anti-SLAMF7-monoclonal antibody (elotuzumab)-based					3 (3.7)
IMiD-based therapies in total	39	8			52 (63.4)
Thalidomide	1	0			1 (1.2)
Lenalidomide	32	8			45 (54.9)
Pomalidomide	6	0			6 (7.3)
Proteasome inhibitors in total	13	2			16 (19.5)
Bortezomib-based	0	1			2 (2.4)
Carfilzomib-based	13	1			14 (17.1)
No current therapy	7	6			13 (15.8)
Therapy lines, median number of therapy lines in total (range)					1 (0–10)
untreated					6 (7.3)
1st line, *n* (%)					42 (51.8)
2nd line, *n* (%)					11 (13.4)
3rd line, *n* (%)					12 (14.6)
>3rd line, *n* (%)					11 (13.4)
Remission status at time of vaccination					
Well-controlled disease	38	12	7		57 (69.5)
Complete remission					16 (19.5)
Very good partial response					41 (50)
Insufficiently controlled disease	14	2	0		16 (19.5)
Partial remission					7 (8.5)
Stable disease					3 (3.7)
Progressive disease					6 (7.3)
Not assessable *					9 (10.9)
TOTAL ^§^	57	17	8		82

MM, multiple myeloma; MGCS, monoclonal gammopathy of clinical significance; MGUS, monoclonal gammopathy of unknown significance; AL, systemic light chain amyloidosis; R/R, relapsed or refractory; BCMA, B-cell maturation antigen; ADC, antibody-conjugate; IMiD, immunomodulatory drugs. * Includes patients newly diagnosed before receiving any treatment or missing data for evaluation or monoclonal gammopathy of clinical significance. ^§^ *n* = 8 individuals with missing antibody evaluation.

**Table 2 cancers-13-03800-t002:** Patient characteristics regarding distribution of lymphocytes and anti-SARS-CoV-2 vaccination.

Variable	Total
Type of anti-SARS-CoV-2 vaccine, *n* (%)	
mRNA-based vaccine	63 (76.8)
Vector-based vaccine	19 (23.2)
Time between first vaccination and Anti-SARS-CoV-2 spike protein titer measurement, days (mean, ± SD)	25.2 (±11.8)
Distribution of lymphocytes detected by flow cytometry	
Mean CD19+ cells/µL (range)	88.5 (1–642)
Mean CD4+ cells/µL (range)	442.7 (39–1730)
Mean CD8+ cells/µL (range)	553.4 (27–2261)

**Table 3 cancers-13-03800-t003:** Effect of CD19+ B-cell count and other covariables on antibody titer values. Results of linear regression model (§).

Variables	All Observations		Observations with Positive Antibody Titer	
	Estimate	*p*-Value		
Intercept	2.86	0.088	4.58	0.068
Log (CD19 count) L	0.43	0.001	0.39	0.008
Time since first vaccination (days)	0.049	0.008	0.022	0.21
Age at vaccination (years)	−0.047	0.027	−0.045	0.03
Insufficiently controlled disease	−1.18	0.024	−0.50	0.47

§ linear regression model: log(antibody titer + 1) = α + β_1_ log(CD19 count) + β_2_ time since vaccination + β_3_ age + ε.

## Data Availability

The data presented in this study are available on request from the corresponding author. The data are not publicly available due to local legal requirements.

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
