# Peer review of "Post-Vaccination Anti-SARS-CoV-2-Antibody Response in Patients with Multiple Myeloma Correlates with Low CD19+ B-Lymphocyte Count and Anti-CD38 Treatment"

_cancers, 2021, doi:10.3390/cancers13153800_

Round 1
Reviewer 1 Report
Ghandili et al., conducted an observational study for patients with hematological malignancies who had received as least one shot of the COVID-19 vaccine. They found that only 23% of patient’s anti-SARS-CoV-2 nAb can be detected. They further found that the CD19+ B lymphocyte counts and patients with anti-CD38 antibody treatment correlated with the nAb titers. This study could provide valuable information, however, this study has not provided details of the study design and statistics descriptions, and exhibited relatively weak rational for the connection of each result.
Major comments:
- Please provide a study flow chart. Flow chart is important to understand the overall idea of this study and help the reader to judge the study design.
- Table 1 should provide the p-value of the variables compared with nAb negative and positive patients, which is the main comparison in this study. It is important to understand the demographic differences between the study groups. The clinical characteristics should be addressed in one table or merged to Table 1, depending on the authors, and the p-value should be provided as well.
- Figure 1 is confusing. Author should show the data using a Table instead of a Figure to address the risk of patients who did not successfully produce nAb. The subhead of the X axis does not reflect the meaning of graphic showed, the “titer” should be a value not just positive/negative. However, author should change the figure to a table.
- The multivariate analysis after univariate analysis should be provided in nAb positive and negative groups.
- It is confusing why certain variables were selected for multiple linear regression in Table 3. If the variables are selected from previous results (ex. table and figure), previous analysis were conducted using independent categorical variable (positive/negative) not continuous variable (titer as shown in Table 3). Author needs to clarify how the analysis of Table 3 was performed and why only few factors selected for multiple linear regression. However, I believed that understanding the association between risk factors and nAb titers would provide more information in this study instead of just addressing the positive/negative.
- If the correlation analysis is the main findings in this study, I would suggest the author provide the graphics for correlation analysis (ex. nAb titer v.s. CD19+ B lymphocyte counts).
Minor comments:
- Since there is only need for one shot for Ad26.COV2.S (Janssen-Cilag International), is it possible for the author to compare the patients who received Ad26.COV2.S with others? That will be interested to understand whether fully vaccinated patients (Ad26.COV2.S) has higher or more numbers of positive of nAb.
- Please remove the result description in the figure legend (Figure 1, lines 208-211).
- In lines 214-217, as Figure 1 is not multivariable linear regression, the paragraph need rewrite.
Author Response
„Please provide a study flow chart. Flow chart is important to understand the overall idea of this study and help the reader to judge the study design.“
- A flow chart was added.
„Table 1 should provide the p-value of the variables compared with nAb negative and positive patients, which is the main comparison in this study. It is important to understand the demographic differences between the study groups. The clinical characteristics should be addressed in one table or merged to Table 1, depending on the authors, and the p-value should be provided as well.”
- The table has been re-structured according to the suggestion of the reviewer
“Figure 1 is confusing. Author should show the data using a Table instead of a Figure to address the risk of patients who did not successfully produce nAb. The subhead of the X axis does not reflect the meaning of graphic showed, the “titer” should be a value not just positive/negative. However, author should change the figure to a table.”
A table was added to the figure giving the numbers of patients producing or not producing antibodies. Titer was judged “positive” or “negative” according to manufacturer’s recommendations; with the high percentage of patient’s not producing antibodies, we see it important to state the number/percentage of those wo do produce antibody titers high enough to be counted as “positive”. As we find it helpful to see this depicted in a figure and the other reviewer also agrees with the figure, we decided not to remove the figure but instead add the above mentioned table to it.
„The multivariate analysis after univariate analysis should be provided in nAb positive and negative groups.”
- We added the linear regression analysis when using nAb positive only. In addition, we calculated a logistic regression model where nAb positive individuals were compared with nAb negative.
“It is confusing why certain variables were selected for multiple linear regression in Table 3. If the variables are selected from previous results (ex. table and figure), previous analysis were conducted using independent categorical variable (positive/negative) not continuous variable (titer as shown in Table 3). Author needs to clarify how the analysis of Table 3 was performed and why only few factors selected for multiple linear regression. However, I believed that understanding the association between risk factors and nAb titers would provide more information in this study instead of just addressing the positive/negative.”
- We used a backward elimination procedure. We now added information on the model selection. In order to facilitate the comparison between the different regression results, we kept the model obtained from the linear regression, total study group and used this model for the restricted dataset as well as for the logistic regression.
„If the correlation analysis is the main findings in this study, I would suggest the author provide the graphics for correlation analysis (ex. nAb titer v.s. CD19+ B lymphocyte counts).”
- Since a high percentage of patients does not produce nAb titers after this first vaccination, a graphic analysis with more than half of the data being nAb titer of 0, the graphical correlation analysis would show lots of titers on the axis and was thus not chosen. However, since we are planning further analyses after the second vaccination, your suggestion of graphical correlation analysis will then be considered.
„Since there is only need for one shot for Ad26.COV2.S (Janssen-Cilag International), is it possible for the author to compare the patients who received Ad26.COV2.S with others? That will be interested to understand whether fully vaccinated patients (Ad26.COV2.S) has higher or more numbers of positive of nAb.”
- Since no included patient received Ad26.COV2.S by Janssen-Cilag International, we chose to delete this sentence.
“Please remove the result description in the figure legend (Figure 1, lines 208-211).”
- The result description was deleted in figure legend 1.
“In lines 214-217, as Figure 1 is not multivariable linear regression, the paragraph need rewrite.”
- “Multivariable” was deleted.

Reviewer 2 Report
The paper by Ghandili at al. presents the data of anti-SarsCoV2 antibody titration after the first dose of vaccination in a cohort of patients with multiple myeloma and related plasma cell neoplasia. The authors also analyze how CD19+ B lymphocytes count, disease status, treatment regimen, age and type of vaccine used influence antibody response. The authors conclude that antibody response mainly correlates with CD19+ B lymphocyte count and a cut-off of 30 lymphocytes/ul is indicated as a possible predictor of antibody response.
The paper is well structured, and the English language is correct with no major issues nor mistakes. The topic presented is highly relevant as anti-COVID19 vaccination is the major weapon now available against the pandemic. Indeed, data obtained from fragile patients are much needed to improve the prevention of the disease in those whose immune system is compromised and therefore need an effective protection against the virus. However, there are some points that may be improved in order to make the manuscript completer and more valuable.
Major points:
- The authors analyze the variability of response to anti-SarsCoV2 vaccination in a cohort of immune compromised patients that has already been reported to have a weaker antibody response. The variability of response is then correlated to several factors. However, no group of healthy controls is present in the study. It may be useful to discuss the variability in response also in the general population comparing the data now available in literature with that collected in the present study.
- The paper claims to be a real-world experience, which is true. However, it is not fully discussed why titration has been done after the first dose and not after the second dose, which is an important booster and may induce immunization in those cases in which the first dose has not been enough. Indeed, the administration of the second dose could potentially change the results significantly.
- The authors also highlight that the timing of titration may be a confounding factor as some patients’ antibodies were titrated only 7 days after vaccination. It should be better explained what the proportion of patients is whose titration was done before a 2-weeks period, considered the minimum for an assessable antibody production.
Minor points:
- Figure 1 reports the number of patients developing antibodies or not developing antibodies and classify them depending on different variables. Although correct, the authors may consider expressing the same data as percentages, thus making the information more easily readable (i.e., among the anti-CD38 mAb treated or not treated patients, what is the percentage of responder and non-responder?).
- Line 236: is the value 0 BAU/ml correct?
- Figure 2: is the Y axis in a Log scale? If so, it should be indicated in the figure.
Author Response
Reviewer 2:
“The authors analyze the variability of response to anti-SarsCoV2 vaccination in a cohort of immune compromised patients that has already been reported to have a weaker antibody response. The variability of response is then correlated to several factors. However, no group of healthy controls is present in the study. It may be useful to discuss the variability in response also in the general population comparing the data now available in literature with that collected in the present study.”
- Additional data on antibody response after the first vaccination in healthy individuals (27,28) and e.g. age as a factor associated with lower antibody titers also in healthy individuals were added in the discussion (29,30).
“The paper claims to be a real-world experience, which is true. However, it is not fully discussed why titration has been done after the first dose and not after the second dose, which is an important booster and may induce immunization in those cases in which the first dose has not been enough. Indeed, the administration of the second dose could potentially change the results significantly.”
- Due to media spreading the word about good immune response already after first vaccination (in healthy individuals), lots of patients assume a similar protection after their first shot.
- Our project is planned as a longitudinal analysis also assessing antibody response after the second vaccination and further data on these will follow. However, we decided to already go forward with publishing these results, as we find it important to enable clinicians to guide and advice patients after the first vaccination to hold up protective measures and to highlight the importance of the second shot. In addition, due to decision of the authorities to prolong intervals between first and second vaccination to enable more first vaccinations, most of the patients had not yet received theirs second vaccination. Given our here presented data, the decision to prolong the time interval may be beneficial for the whole population but not for patients with plasma cell neoplasia on an individual basis.
“The authors also highlight that the timing of titration may be a confounding factor as some patients’ antibodies were titrated only 7 days after vaccination. It should be better explained what the proportion of patients is whose titration was done before a 2-weeks period, considered the minimum for an assessable antibody production.”
- Due to the real-world setting anti-SARS-CoV-2 Spike protein titers were tested before day 14 in 11 of 74 assessable patients. This information was added to the limitations part in the manuscript. Given the potential hazard of additional clinic visits, we did not deem it ethical to invite patients for extra appointments with the only intention to assess their antibody titer at a “better” timepoint, although we are fully aware that this is a potential confounding factor.
“Figure 1 reports the number of patients developing antibodies or not developing antibodies and classify them depending on different variables. Although correct, the authors may consider expressing the same data as percentages, thus making the information more easily readable (i.e., among the anti-CD38 mAb treated or not treated patients, what is the percentage of responder and non-responder?).”
- Table with numbers and percentages added to the figure.
“Line 236: is the value 0 BAU/ml correct?”
- A median of 0 BAU/ml is correct indicating that more than 50% of individuals did not develop any antibodies.
“Figure 2: is the Y axis in a Log scale? If so, it should be indicated in the figure.”
- Yes, it is in Log scale. This information was added.

Round 2
Reviewer 1 Report
The authors have addressed my comments appropriately and the manuscript can be accepted in present form.